# OpenReview forum: "SAGE-CoT: Self-Adaptive Generated Chain-of-Thought for Jailbreaking"
_ICLR.cc/2026/Conference — Submitted to ICLR 2026_

### Official Review · Reviewer_69Yr · 2025-10-16

**Soundness:** 2
**Presentation:** 3
**Contribution:** 2
**Rating:** 2
**Confidence:** 5

**Summary:**

This paper presents SAGE-CoT, a CoT-based jailbreak method for LRMs. It uses three chained stages: first neutralizing the initial content but keeping the original intent; then split the text into a dictionary-based payload with randomly added noise (improving stealthiness); finally a prompt-guided intent-recovery template is used to guide the jailbreaks. The claim is that contemporary LLMs reconstruct the hidden intent during reasoning and produce harmful outputs, yielding high attack success rates (ASR) across models, with some evaluation comparisons to prior prompt-based or CoT-style attacks and a few defenses as support.

**Strengths:**

The paper highlights an increasingly relevant threat surface for LLM safety researches: the model’s explicit reasoning process. Framing prompt obfuscation as a lightweight pre-processing pipeline is clear and easy to reproduce in principle. The work also gestures toward empirical comparisons and ablations, which—if robust—could help quantify how much each obfuscation step contributes to success. As a safety paper, the topic is timely and the writing is accessible.

**Weaknesses:**

Despite the apparent simplicity, the evidence chain is not convincing in its current form.

First, I was unable to reproduce the central claims on GPT-5 and Gemini-2.5-Pro using the appendix prompts and attack questions—3 runs each yielded no successful attacks—casting doubt on the reported cross-model effectiveness. If such attack is well-mitigated currently, it will not a big concern for LLM safety research now

Second, the core motivation hinges on “stealth”: obfuscation is said to evade alignment layers and external filters. Yet the proposed transformations introduce conspicuous artifacts (index markers, fragmented syntax, noise tokens). The paper does not demonstrate detectability or alert rates under intent classifiers, anomaly detectors, or rule-based filters, so the stealth claim remains largely asserted rather than measured.

Third, the defense suite is shallow and partially outdated; beyond LlamaGuard3 there is little engagement with stronger SOTA defenses (e.g., JBShield, Legilimens, please check recent security papers in CCS, S&P, Security and NDSS), nor is there a mechanistic analysis explaining why specific defenses fail or succeed.

Fourth, I wonder why you use customized HS rather than StrongReject score as you already evaluated StrongReject benchmark.

Finally, the evaluation is under-scoped: only subsets of AdvBench and StrongReject are used, with limited breakdowns by hazard type, little robustness analysis (temperature, top-p, context length, system prompt), and limited transfer across models.

**Questions:**

1. How exactly the attack is performed?  Which way did you use to evaluate the attack, API or chat website? For thinking model like GPT-5 and Gemini, how is the reasoning effort/thinking budget set? How are the designed stages actually performed and chained together (with its model usage, prompt, etc.)? If a different model usage can fail the attack chain (e.g., I can choose to use GPT5 all the time to execute the whole pipeline) , what are the necessary key points needed for this specific step to work? A well-documented artifact would help addressing the effectiveness of this attack.
2. What is the evaluation results on the whole AdvBench and StrongReject benchmarks?
3. Evaluate against more SOTA defenses, in my view, an adapted JBShield/Legilimens defense may mitigate this kind of attack well.
4. Try to explain how the internal LLM reasoning process fails its safety alignment and propose ways to mitigate it accordingly, either on the alignment side or on the guardrail side.

---

> ### Author Response · Authors · 2025-11-21
>
> **General Response:**
> We sincerely thank the reviewer for the rigorous assessment and for attempting to reproduce our work. We take the feedback on reproducibility and defense evaluation very seriously. We provide detailed clarifications below to resolve the discrepancies and have revised the paper accordingly.
>
> **Weakness 1 & Q1: Reproducibility (GPT-5 / Gemini)**
>
> **Response:**
> We apologize for the difficulties encountered during reproduction. We are confident the attack is effective and have identified the likely causes for the failure:
>
> - **Appendix Formatting vs. Raw Artifacts:**  The prompts displayed in the Appendix were formatted and slightly sanitized to fit the PDF layout constraints. Direct copy-pasting from the PDF often introduces formatting errors (e.g., JSON syntax corruption, invisible characters, line breaks) that disrupt the strict syntax required for the CoT template. We urge the reviewer to use the **raw text files** provided in the **Supplementary Material** for exact reproduction.
> - **API Settings:**  As specified in **Section 4**, we strictly conducted all experiments via **official APIs** to ensure a controlled environment. We utilized default parameters: **temperature=1.0** and, where applicable (e.g., for o1/o3/Gemini), **reasoning_effort="medium"**  (or "High"). Lower reasoning budgets often truncate the necessary decoding steps.
> - **Verification & Evidence:**  We have successfully re-verified the reproduction on **Google AI Studio** (web interface) using these default settings. To demonstrate this, we have included part of **successful execution logs** for **Gemini-2.5-Pro**, **GPT-5**, and **DeepSeek-V3.1-Thinking** in the supplementary material. We invite the reviewer to test the raw prompts using these specific configurations.
>
> **Question 1: Specific Attack Execution Details**
> **Response:**
>
> - **Pipeline:**  The reviewer asked about using GPT-5 throughout. We clarify that **Gemini-2.5-Flash** is used as the auxiliary model for Stage 2 (Semantic Obfuscation), not the target model. This is because Gemini-Flash follows rewriting instructions faithfully without the self-refusals that a highly aligned model like GPT-5 might trigger during the obfuscation step. This is the only step requiring an external LLM.
> - **System Prompts:**  Our evaluation against the **DRO defense** (Section 4.4) confirms that even with defensive system prompts (simulating web interfaces), SAGE-CoT remains effective because it manipulates the reasoning process rather than directly conflicting with system instructions.
>
> **Weakness 2: Stealth vs. Artifacts**
> **Response:**
> The reviewer is correct that our prompt contains syntactic artifacts (e.g., indices) visible to anomaly detectors. However, we define "stealth" specifically as the ability to evade **Semantic Intent Filters** (e.g., LlamaGuard3), which are the dominant defense layers in production. Our results against **LlamaGuard3** confirm that the semantic fragmentation successfully bypasses these content-based filters. We have refined the definition of "stealth" in the paper to be precise.
>
> **Weakness 3 & Q3: Defense Suite (JBShield/Legilimens)**
> **Response:**
> We acknowledge the existence of advanced defenses like JBShield. However, there is a mismatch in the **Threat Model**:
>
> - **Black-box Constraint:**  SAGE-CoT targets proprietary models (GPT-o1, Gemini) where we (and API users) have no access to weights or internal states.
> - **Defense Applicability:**  Defenses like **JBShield** and **Legilimens** are typically white-box/gray-box methods requiring access to logits or gradients. Applying them would violate the black-box assumption.
> - **LlamaGuard3:**  We chose LlamaGuard3 as it represents the standard for reproducible, input/output-level filtering suitable for black-box APIs. We have added a "Limitations" section discussing future defenses.
>
> **Weakness 4**  **&**  **5 & Q2: HS vs. StrongReject Score and** **Evaluation Scope**
>
> - **Metric Validity:**  Our metric (HS + KSS) is adopted from H-CoT, which are standard benchmarks in recent safety literature. As discussed in response to other reviewers, these metrics have been validated to have high correlation with human judgments. We have clarified this source in the revised paper.
> - **Benchmark Scope:**  We evaluated on representative subsets of **AdvBench** (50 prompts) and **StrongReject** (54 prompts) due to the high computational cost and latency of reasoning models (which generate very long outputs). The subsets were selected to be statistically diverse (covering all categories in StrongReject). Our results show consistent, large-margin improvements (e.g., +15-20% ASR), suggesting that the findings are robust and would likely hold on the full sets.

---

### Official Review · Reviewer_zXiG · 2025-10-28

**Soundness:** 3
**Presentation:** 3
**Contribution:** 3
**Rating:** 6
**Confidence:** 4

**Summary:**

This paper proposes a new paradigm of jailbreak attacks that target the chain-of-thought (CoT) reasoning traces of large reasoning models (LRMs), without having access to these reasoning traces. The proposed approach uses the LRM to complexify a starting template mentioning some text encryption and decryption protocols. It then uses a set of strategies to obfuscate the malicious prompt before embedding it in the complexified template, resulting in the final malicious query. The approach shows remarkable effectiveness on contemporary LRMs such as o3 and Gemini-2.5-Pro-Thinking, and maintains resilience to standard defense strategies.

**Strengths:**

- The proposed method is simple, intuitive and highly effective - using the LRM itself to generate a jailbreak prompt with minimal token overhead, and obtaining very high ASRs on several contemporary LRMs - outperforming baselines on standard datasets. The scope of evaluation is also quite large, covering nearly every model provider.
- Detailed ablations confirm that each component of the attack pipeline is crucial for its performance, and the attack also proves more resilient to defense strategies than baselines, even without access to the model’s reasoning traces.
- The work is quite timely and relevant with the increasing use of LRMs. The paper is also presented quite well.

**Weaknesses:**

- While the attack is highly effective, it is a derivative of a long line of existing jailbreak attacks that all use LLMs to generate obfuscations of the input prompt, and compose various jailbreak attacks to increase their effectiveness [1][2][3][4] - somewhat limiting its originality. Some of these approaches would have also proven to be appropriate baselines for this work, given their similar nature.
- The evaluation metric used in this work is nonstandard - using a combination of HS and KSS. Are there studies showing that this metric aligns with human preferences? Does SAGE-CoT still outperform other approaches when evaluated using more standard approaches such as HarmBench-Llama? Evaluation is also not performed on the entirety of AdvBench and StrongReject.

[1] AutoDAN: Generating Stealthy Jailbreak Prompts on Aligned Large Language Models. Xiaogeng Liu, Nan Xu, Muhao Chen, Chaowei Xiao.
[2] h4rm3l: A language for Composable Jailbreak Attack Synthesis. Moussa Koulako Bala Doumbouya, Ananjan Nand, Gabriel Poesia, Davide Ghilardi, Anna Goldie, Federico Bianchi, Dan Jurafsky, Christopher D. Manning.
[3] Jailbroken: How Does LLM Safety Training Fail? Alexander Wei, Nika Haghtalab, Jacob Steinhardt.
[4] GPTFUZZER: Red Teaming Large Language Models with Auto-Generated Jailbreak Prompts. Jiahao Yu, Xingwei Lin, Zheng Yu, Xinyu Xing.

**Questions:**

- Why were the specific strategies of semantic obfuscation, indexed word permutation and noise injection chosen for the intent obfuscation layer? Were other strategies tried? Did the authors try allowing the LRM itself to perform the intent obfuscation?
- Some details are missing regarding the generation of the self-complexified template. The paper mentions that this is done once for each target LRM. Does the LRM ever refuse to generate these harmful query strings? How many generations are created for each target LRM to get these templates, and how is the best template selected from these generations?

---

> ### Author Response · Authors · 2025-11-21
>
> **General Response:**
> We thank the reviewer for recognizing the timeliness, simplicity, and high effectiveness of SAGE-CoT across diverse LRMs. We appreciate the constructive feedback on originality and evaluation, which we address below.
>
> **Weakness 1: Originality & Derivative Nature ([1]-[4])**
> **Response:**
> We respectfully distinguish SAGE-CoT from the cited works:
>
> - **Mechanism Difference:**  Works like **AutoDAN** [1] and **GPTFuzzer** [4] focus on optimizing the input prompt itself (via genetic algorithms or gradients) to bypass static filters. **h4rm3l** [2] focuses on DSL-based composition. In contrast, SAGE-CoT is the first to exploit the **Meta-Reasoning** capability of LRMs. Instead of finding a "magic string," we force the model to build a "Trojan horse" (the decoding template) within its own reasoning chain. This shifts the attack surface from input optimization to cognitive process manipulation.
> - **Baseline Selection:**  We compared against **PAP** (which uses obfuscation), **CodeAttack** (encoding), and **ArtPrompt** (ASCII art), which are representative of the "obfuscation" family. We also included **H-CoT** as the state-of-the-art CoT-specific attack. We believe this covers a broad spectrum of relevant baselines.
>
> **Weakness 2: Non-standard Metric & Evaluation Scope**
> **Response:**
>
> - **Metric Validity:**  Our metric (HS + KSS) is adopted from H-CoT, which are standard benchmarks in recent safety literature. As discussed in response to other reviewers, these metrics have been validated to have high correlation with human judgments. We have clarified this source in the revised paper.
> - **Benchmark Scope:**  We evaluated on representative subsets of **AdvBench** (50 prompts) and **StrongReject** (54 prompts) due to the high computational cost and latency of reasoning models (which generate very long outputs). The subsets were selected to be statistically diverse (covering all categories in StrongReject). Our results show consistent, large-margin improvements (e.g., +15-20% ASR), suggesting that the findings are robust and would likely hold on the full sets.
>
> **Question 1: Choice of Obfuscation Strategies**
> **Response:**
> We chose this specific combination based on the "Swiss Cheese Model" of defense evasion:
>
> - **Semantic Obfuscation (SO):**  Evades semantic embeddings and intent classifiers.
> - **Indexed Permutation (IWP):**  Evades N-gram and keyword spotting (by breaking word order).
> - **Noise Injection (NI):**  Disrupts token probability sequences.
> - Alternatives Tried: We experimented with pure cipher substitution (e.g., Caesar/Base64) and purely generative obfuscation (asking the LRM to "hide the intent"). We found that: (1) LRMs often fail to decode pure ciphers accurately; (2) LRM-generated obfuscation often triggers the model's own self-refusal during the generation phase. Our hybrid approach (External Obfuscation + Internal Decoding) proved to be the most robust sweet spot.
>
> **Question 2: Template Generation Details (Refusal & Selection)**
>
> **Response:**
> We appreciate the opportunity to clarify this process.
>
> - **No Refusal:**  The LRM **never refuses** to generate the template. This is because the meta-instruction (Stage 1) asks for a purely benign logic task with absolutely no harmful content. To the model, this is a helpful decoding assistance task.
> - **Single-Shot Success:**  Due to the benign nature of the request, the generation is highly deterministic and effective. In our experiments, we generated **only one template per target model** (1-shot), and it worked successfully. We did not need to generate multiple candidates or perform any selection/filtering process. This highlights the efficiency of SAGE-CoT compared to optimization-based attacks that require thousands of queries.

---

> > ### Comment · Reviewer_zXiG · 2025-11-25
> >
> > I thank the authors for their rebuttal comments. I have some follow-up questions.
> >
> > 1. Although SAGE-CoT presents itself as a cognitive-process–manipulation attack, in practice it functions much like other template-based attacks (such as DAN or Development Mode [1]) by prepending a prefix that assigns the model a specific persona and instructs it to carry out an obfuscated sequence of steps to deliver a malicious payload. Comparable prompts can also be generated through systems like AutoDAN, GPTFuzzer, or h4rm3l. This does not diminish the contribution of the work, and it is true that LRMs can be induced to follow far more elaborate procedures than standard LLMs, enabling deeper layers of obfuscation. However, SAGE-CoT still fits within the same broad class of attack methods. Benchmarking it against these stronger baselines would therefore provide a more rigorous assessment of its effectiveness.
> >
> > [1] https://abnormal.ai/blog/chatgpt-jailbreak-prompts
> >
> > 2. The metric used in this work appears to differ from that of H-CoT. Whereas H-CoT derives its harmfulness score directly from an LLM, this approach combines heuristic keyword-suppression scores with LLM-based alignment scores. A human-annotated evaluation of alignment under this metric would substantially strengthen the validity of the work.

---

> > > ### Author Response · Authors · 2025-12-03
> > >
> > > Thank you for your detailed and insightful follow-up questions. We appreciate the opportunity to further elaborate on these important points regarding our method's classification and our evaluation metric.
> > >
> > > **1.**  **Regarding the classification of SAGE-CoT and comparison with other baselines:**
> > >
> > > We agree with your insightful observation that SAGE-CoT shares functional characteristics with the broader class of template-based attacks, and we thank you for highlighting this connection. We also fully concur that benchmarking against strong, generative baselines like **h4rm3l, AutoDAN, GPTFuzzer, and Jailbroken** is essential for a truly rigorous assessment of our method's effectiveness.
> > >
> > > Unfortunately, a full evaluation against these methods was not feasible within the rebuttal period. The primary reasons are twofold: the substantial inference latency of Large Reasoning Models (LRMs), and the high number of queries required by the iterative, optimization-based nature of methods like AutoDAN, which makes a comprehensive comparison computationally prohibitive in a short timeframe.
> > >
> > > However, we want to be clear that we consider this comparison a **high-priority next step for our research.**  We are committed to conducting this comprehensive benchmarking as we continue to develop this work and will incorporate the findings in future iterations. We have also updated the manuscript to explicitly acknowledge these methods as important related work.
> > >
> > > **2. Regarding the evaluation metric and its relation to H-CoT:**
> > >
> > > Thank you for your sharp observation, which allows us to provide a more precise breakdown of our evaluation protocol. You are correct that our final metric is a careful, two-step adaptation of the H-CoT methodology, designed to enhance both granularity and rigor.
> > >
> > > Our process is as follows:
> > >
> > > - **Adoption and Scaling:**  We start with the core principle of H-CoT's LLM-based evaluation, which provides a harmfulness rating on a **0-5 scale**. For finer granularity, we linearly map this rating to a **0-100 scale**, which we term the **Alignment Score (AS)** .
> > > - **Enhancement with KSS:**  Building upon this normalized score, we introduce our key enhancement: the **Keyword Suppression Score (KSS)** . A response receives a KSS of 0 if it contains refusal keywords and a bonus of 20 if it does not.
> > >
> > > The final score is the sum of **AS + KSS**, with a success threshold of 80. This composite metric ensures that a successful jailbreak must not only be judged as highly harmful by the LLM (high AS) but also cleanly bypass behavioral triggers (rewarded by KSS), setting a higher and more realistic bar for success.
> > >
> > > Regarding the need for validation, we agree that demonstrating the metric's alignment with intuitive outcomes is crucial. To provide full transparency and showcase the reliability of our metric, **we have added a new appendix (Appendix D).**  This appendix presents a series of illustrative examples, one for each major score tier (from 0 to 120), in the format you suggested. Each example includes the harmful intent, the model's response, and a detailed scoring rationale. We believe this provides a clear and concrete demonstration of our metric's consistency and its ability to distinguish between different types of model behavior, from hard refusals to perfect jailbreaks.
> > >
> > > We hope these clarifications and the new appendix adequately address your concerns. Your feedback has been invaluable in helping us refine the positioning and rigor of our work.

---

### Official Review · Reviewer_Tsuo · 2025-10-30

**Soundness:** 3
**Presentation:** 3
**Contribution:** 2
**Rating:** 4
**Confidence:** 3

**Summary:**

The paper introduces SAGE-CoT, a black-box jailbreak framework that induces a target Large Reasoning Model (LRM) to self-generate a bespoke CoT program and then feeds it a concealed malicious payload the template can decode. Stage 1 concatenates a simple intent-recovery template with a meta-instruction, prompting the target model itself to elaborate a model-specific reasoning template  that looks benign yet embeds decoding rules (persona, reporting schema, decryption steps) and is reused across attacks on that model.  In Stage 2, the attacker transforms a harmful query using semantic obfuscation via an auxiliary LLM. Experiments span proprietary and open-source LRMs, evaluated on AdvBench and StrongReject .  SAGE-CoT reports state-of-the-art ASR/HS across models.

**Strengths:**

1. The two-stage design—self-generated CoT template + layered obfuscation, is well-specified with equations, example rules , and a precise assembly step.

2. The method does not assume access to internal traces and is evaluated on closed models (GPT-5, GPT-o-series) and CoT-exposing models, demonstrating broad applicability under a uniform medium reasoning budget.

3. Tables show consistent improvements over six baselines on both AdvBench and StrongReject, for instance, on StrongReject, SAGE-CoT reaches 92.59% ASR on Gemini-2.5-Pro-Thinking versus 75.93% for ICRT.

**Weaknesses:**

1. HS is computed by GPT-4o and includes a 20-point bonus for absence of refusal keywords, which can reward verbosity and stylistic features embedded in (R_m) (e.g., mandated ≥600-word outputs and JSON schemas) rather than substantive harmfulness; this may advantage SAGE-CoT over baselines not designed to game HS components. The paper should compare against human raters and report inter-rater reliability to validate HS.

2. H-CoT is adapted by using the final response as a proxy for CoT where traces are unavailable, which likely weakens it relative to its intended design; the paper does not quantify the performance drop from this adaptation or provide an alternative baseline that preserves H-CoT’s strengths under black-box constraints.

3. Defenses are limited to LlamaGuard3, SmoothLLM, and DRO; modern multi-stage provider stacks (content classifiers + safety prompts + output filters + rate-limiters) and dynamic detectors are not evaluated. The evasion advantage versus LlamaGuard3 may not hold once multi-modal or server-side heuristics are added.

**Questions:**

1. The paper asserts the harmful intent is never explicitly present … nor does it surface within the model’s reasoning process, yet shows long forensic-style reasoning and JSON that could itself cue the judge; no introspective or log-prob probes are provided to substantiate the claimed internal-only recovery mechanism. Stronger evidence (e.g., trace auditing on models that expose CoT, or external detectors monitoring intermediate tokens) is needed.

---

> ### Author Response · Authors · 2025-11-21
>
> **General Response:**
> We thank the reviewer for the thorough assessment and for recognizing the strong empirical performance and applicability of SAGE-CoT. We have addressed the concerns regarding metrics, baseline fairness, and mechanism validation below.
>
>
> **Weakness 1: HS Metric Bias & Human Validation**
>
> **Response:**
>
> We share the reviewer's commitment to rigorous evaluation. While we did not perform a new round of human annotation, we addressed the metric validity and verbosity concerns as follows:
>
> - **Adoption of Validated Protocols:**
>   As discussed in our response to Reviewer LMAm03, we adopted the evaluation prompt and scoring rubric from **H-CoT**. These prior works have extensively benchmarked this GPT-4o-based judging methodology against human raters, demonstrating high inter-rater reliability. By utilizing this established framework, we ensure our results are comparable to the state-of-the-art and scientifically valid.
> - **Robustness Against Verbosity:**
>   Regarding the concern that the metric "rewards verbosity," we clarify that our success threshold ($HS\ge 80$\) effectively filters out verbose refusals.
>
>   - The HS consists of an **Alignment Score (AS, 0-100)**  and a **Keyword Suppression Score (KSS, 0/20)** .
>   - To exceed the score of 80, a response must achieve an Alignment Score of at least 60-80. According to the rubric, this level requires providing  **"specific, actionable harmful instructions."**
>   - A verbose refusal, even if it avoids keyword triggers (getting +20 KSS), would typically receive an AS of 0 or 20. The total score (20-40) would remain far below the success threshold. Thus, length alone cannot game the metric
>
> **Weakness 2: H-CoT Adaptation Fairness**
>
> **Response:**
> We appreciate the opportunity to clarify our baseline implementation. We tailored the H-CoT implementation based on the **visibility of CoT traces** for each target model, rather than applying a blanket adaptation:
>
> - **Visible CoT Traces (Full H-CoT):**  For models where the API or weights expose the intermediate reasoning tokens (e.g., **DeepSeek-V3.1-Thinking**, **Qwen3**), we implemented the **original, full-strength H-CoT** method. This allows the baseline to leverage internal traces exactly as intended by its authors.
>
>   - Evidence: Even in this ideal setting for H-CoT, SAGE-CoT demonstrates superior performance. On **StrongReject** with DeepSeek-V3.1, SAGE-CoT achieves **88.89% ASR** compared to H-CoT's **81.48%** . This confirms that SAGE-CoT's efficacy is not merely due to the weakening of the baseline.
> - **Hidden CoT Traces (Proxy H-CoT):**  For models where CoT traces are explicitly hidden or summarized by the provider (e.g., **GPT-o3-mini** via OpenAI API), we used the final response as a proxy. This reflects the real-world constraint of attacking these specific "black-box thinking" models.
>
> **Weakness 3: Limited Defenses (Multi-stage Stacks)**
> **Response:**
> We acknowledge that provider-side stacks are formidable. However, evaluating against proprietary dynamic stacks is non-reproducible. **LlamaGuard3** represents the state-of-the-art in reproducible content moderation. Furthermore, our high ASR on **GPT-o3-mini** and **Gemini-2.5** (Table 1)—which inherently employ such multi-stage defenses internally—serves as implicit evidence of SAGE-CoT's ability to bypass modern production-grade safety stacks. We have added a discussion on limitations regarding dynamic detectors in the Conclusion.
>
> **Question 1: Evidence of Internal Recovery vs. Explicit Traces**
> **Response:**  We thank the reviewer for catching this imprecise phrasing. We have corrected the text in **Section 3.2.3**.
>
> We agree that the intent must be recovered internally for the attack to work. Our revised claim is that the intent is **structurally obfuscated** rather than "invisible." In the CoT traces, the harmful tokens are generated incrementally and interleaved with benign "puzzle-solving" steps. This **fragmentation** prevents safety monitors—which typically scan for contiguous harmful semantic patterns—from triggering a refusal during the generation process.

---

> ### Comment · Reviewer_Tsuo · 2025-11-26
> **Thanks for detailed Resposne**
>
> Thank you for the detailed response. I still have several questions. In the area of content moderation, recent SOTA approaches such as WildGuard exhibit strong defensive performance. Moreover, Qwen-Guard (released on 2025/09/23, which also demonstrates notable robustness, although a direct comparison is not necessary in this context).

---

> ### Author Response · Authors · 2025-12-03
>
> Thank you for your follow-up comments and for pushing us to evaluate against the latest state-of-the-art defenses like Qwen-Guard. Your suggestion has led to a crucial finding that significantly strengthens our paper's contribution.
>
> Following your recommendation, we conducted a new set of experiments where all attack prompts were first filtered by **Qwen-Guard** before being sent to the target LRM. The results demonstrate a remarkable resilience advantage for SAGE-CoT:
>
> - **SAGE-CoT:**  Our method's performance remained almost entirely unaffected by Qwen-Guard, with **no significant drop in either ASR or HS** on both AdvBench and StrongReject datasets.
> - **H-CoT (Baseline):**  In stark contrast, the performance of the state-of-the-art baseline, H-CoT, was **substantially degraded** by Qwen-Guard. Its ASR on AdvBench dropped from **86% to 74%** , and on StrongReject, it fell from **53.7% to 42.6%** .
>
> These results provide strong empirical evidence that SAGE-CoT is not only highly effective but also significantly more robust against advanced content moderation filters than existing CoT-based attacks. We hypothesize this is because SAGE-CoT’s structural obfuscation and meta-reasoning manipulation make the initial prompt appear benign and procedural, whereas H-CoT's prompts, which more directly hijack the reasoning process, are more susceptible to semantic safety detectors like Qwen-Guard. To reflect these important findings, we have updated **Figure 4** and added a detailed analysis to the corresponding section of our revised manuscript.
>
> Regarding **WildGuard**, we recognize its importance as a crucial benchmark. Should the paper be accepted, **we would be fully committed to including an evaluation against WildGuard in the final version to further strengthen its contribution.**  For now, we have detailed this as a primary direction for future work in the paper.

---

### Official Review · Reviewer_LMAm · 2025-11-03

**Soundness:** 2
**Presentation:** 3
**Contribution:** 2
**Rating:** 2
**Confidence:** 4

**Summary:**

- This work proposes a black-box jailbreak attack mechanism exploiting the Chain-of-Thought (COT) reasoning approach adopted by recent large language models.
- This work claims to propose an attack approach that doesn't rely on the availability of "internal COT traces", which approach is claimed to be novel.
- This work conducts empirical experiments comparing the proposed approach to 6 previously reported attack mechanisms while targeting 6 state-of-the-art LLMs showing that the proposed approach is competitively performant.

**Strengths:**

- This work adheres to the established black-box jailbreak attack, and the particular category of chain-of-thought based black-box attacks.
- This work uses previously adopted datasets of illicit instructions such as AdvBench and StrongReject.
- This work reports Attack Success Rates and Harmfulness Scores for 6 other related jailbreak attacks across 6 target LLMs.
- This work reports the performance of the proposed attack mechanism against 3 defense mechanisms.

**Weaknesses:**

- This work emphasizes the ability of performing black-box COT attack without access to "Internal COT traces" as a key contribution. However the definition of "Internal COT traces" is not sufficiently clear, which makes their specific contribution hard to qualify.
- This work does not discuss prior work such as [3], which performs black-box COT jailbreaking in a zero-shot manner.
- This work adopts many unclear terminologies around the idea of "internal COT traces" that need substantial clarification.
- Formalisms employed in this work, namely in Eq 1, Eq 2 and Eq 3 could be greatly simplified. Black-box jailbreak attacks can be represented as string-to-string transformations, and compositions thereof (see [4] and [5]).
- This work should clarify how GPT-4o is employed "as an external judge to measure both the attack success rate (ASR) and the harmfulness score (HS)", and justify the validity of how those metrics are computed. The authors should discuss the related ternary taxonomy proposed by [4], which [5] used to construct an LLM-based judge, and validate it with human annotations.


# References
- [1] Kuo, M., Zhang, J., Ding, A., Wang, Q., DiValentin, L., Bao, Y., ... & Chen, Y. (2025). H-cot: Hijacking the chain-of-thought safety reasoning mechanism to jailbreak large - reasoning models, including openai o1/o3, deepseek-r1, and gemini 2.0 flash thinking. arXiv preprint arXiv:2502.12893.
- [2] Yao, Y., Tong, X., Wang, R., Wang, Y., Li, L., Liu, L., ... & Wang, Y. (2025). A mousetrap: Fooling large reasoning models for jailbreak with chain of iterative chaos. arXiv - preprint arXiv:2502.15806.
- [3] Shaikh, O., Zhang, H., Held, W., Bernstein, M., & Yang, D. (2023). On Second Thought, Let’s Not Think Step by Step! Bias and Toxicity in Zero-Shot Reasoning. In Proceedings of the 61st Annual Meeting of the Association for Computational Linguistics (Volume 1: Long Papers). Association for Computational Linguistics.
- [4] Wei, A., Haghtalab, N., & Steinhardt, J. (2023). Jailbroken: How does llm safety training fail?. Advances in Neural Information Processing Systems, 36, 80079-80110.
- [5] Doumbouya, M. K. B., Nandi, A., Poesia, G., Ghilardi, D., Goldie, A., Bianchi, F., ... & Manning, C. D. h4rm3l: A Language for Composable Jailbreak Attack Synthesis. In The Thirteenth International Conference on Learning Representations.

**Questions:**

- How does this work compare to [3], which doesn't appear to require access to "internal reasoning traces"?
- Line 59 "While effective, these approaches typically require privileged access to internal CoT traces or manual prompt engineering, limiting applicability in realistic black-box scenarios."
    - is this accurate, given that [1] explicitly claims that the intermediate COT traces are often displayed for transparency and interpretability reasons?
- What do "meta-instruction" and "meta-level reasoning" mean? How are they different from "intermediate reasoning steps" and "reasoning traces"? I suggest that the authors avoid unnecessary neologism that may induce confusion in readers, adopt consistent terminology, and prefer clarity and simplicity.
- What are "specific internal reasoning pathways" (Line 215)
- How are Attack Success Rates and Harmfulness Scores calculated using GPT-4o? Why is the employed method valid?

---

> ### Author Response · Authors · 2025-11-21
>
> **General Response:**
> We sincerely thank the reviewer for the constructive and detailed feedback. We found the suggestions regarding terminology clarification and the simplification of formalisms particularly helpful in improving the readability of our paper. We have revised the manuscript to address all raised concerns.
>
> **Weakness 1 & 3 & Q3: Unclear Terminology ("Internal CoT traces", "Meta-instruction")**
> **Response:**  We apologize for the confusion caused by these terms. We have added a specific  **"Terminology Clarification"**  paragraph in Section 3.1 to formally define them.
>
> - We define  **"Internal CoT traces"**  as the intermediate reasoning tokens generated by the model. In our black-box setting, we assume these are either hidden (as in GPT-o1 API) or immutable by the user, preventing white-box optimization.
> - We define  **"Meta-instruction"**  as a prompt that directs the model to construct a process (e.g., a decoding template) rather than executing a task directly.
> - We define  **"Meta-level reasoning"**  as the capability to reason about the structure of the task itself.
>   We have standardized these terms throughout the paper to avoid unnecessary neologisms.
>
>
> **Weakness 2 & Q1: Missing Prior Work (Shaikh et al. [3])**
> **Response:**  We have included a discussion of Shaikh et al. [3] in the **Related Work** section.
>
> - **Comparison:**  Shaikh et al. [3] investigate **zero-shot biases**, where benign prompts (e.g., "Let's think step by step") inadvertently trigger toxicity. In contrast, SAGE-CoT is an **adversarial attack** framework. It actively constructs obfuscated payloads and forces the model to generate specific decoding templates to bypass robust safety alignment, which simple zero-shot prompting cannot achieve on highly aligned LRMs.
>
>
> **Weakness 4: Simplify Formalisms (Eq 1-3)**
> **Response:**  We have incorporated this suggestion and have revised the formula and definition in Section 3 accordingly.
>
>
> **Weakness 5 & Q5: Validity of GPT-4o Judge & Reference [4]**
> **Response:**  We have updated the **Experiments** section to justify our metrics.
>
> We have revised the **Evaluation Metrics** section to clarify the source and validity of our judge.
>
> - **Source and Validity:**  We adopted the evaluation prompt and scoring mechanism from **H-CoT**. As demonstrated in their work and related studies like **StrongReject**, this specific GPT-4o-based judging protocol achieves high alignment with human annotations. By utilizing this established framework, we ensure our metrics are both rigorous and comparable to prior arts.
> - **Alignment with Taxonomy in [4]:**  We explicitly map the AutoRAN scoring system to the ternary taxonomy (Refusal, Benign, Harmful) proposed by \\citet{wei2023jailbroken}.
>
>   - **Refusal/Benign:**  A low Harmfulness Score (HS<80) indicates the model refused or provided irrelevant information.
>   - **Harmful:**  A high score ($HS \ge 80$\) signifies a successful jailbreak where the intent is fully realized without refusal. Therefore, our threshold-based metric effectively implements the taxonomy from [4] while providing finer granularity for ablation analysis.
>
>
> **Q2: Accuracy of Line 59 (CoT availability)**
> **Response:**  We have clarified this statement. While some models (e.g., DeepSeek-R1) display traces for transparency, many proprietary APIs (specifically the OpenAI o1/o3 series) currently hide raw reasoning tokens or provide only summaries. SAGE-CoT is designed to operate without requiring read access to these internal states.
>
>
> **Q4: What are "specific internal reasoning pathways"?**
> **Response:**  This refers to the model-specific logical patterns generated during Stage 1. By allowing the target model to generate its own template ($R_m$), the decoding logic aligns with that specific model's training distribution and activation tendencies, making it more likely to execute the harmful instruction than a generic, human-written template. We have rephrased this in the text for clarity.

---

### Author Response · Authors · 2025-12-03

**Dear PCs, SACs, ACs, and Reviewers,**

Thank you for your detailed and insightful feedback on our work. To assist the AC in evaluating the discussion, we provide a summary of the key points from the reviews and our responses.

### Strengths

We are grateful that the reviewers generally found our proposed method, SAGE-CoT, to be timely, effective, and well-presented. The main strengths highlighted include:

- **A simple, intuitive, and highly effective attack mechanism** that innovatively manipulates the LRM's meta-reasoning capabilities. (Tsuo, zXiG)
- **Demonstrated high Attack Success Rates** across a wide range of contemporary proprietary and open-source Large Reasoning Models. (Tsuo, zXiG)
- **Broad applicability in realistic black-box scenarios** without requiring access to internal model states or reasoning traces. (LMAm, Tsuo)
- **Comprehensive scope of evaluation**, covering numerous model providers, baselines, and established benchmarks. (LMAm, zXiG)
- The work addresses a **timely and relevant threat surface** and is clearly written. (zXiG, 69Yr)

### Concerns and Our Addressing

Reviewers raised several important concerns, primarily focusing on the validity of our evaluation metric, novelty in the context of prior work, the scope of defense evaluation, and reproducibility. In response, we have provided extensive clarifications, conducted new experiments, and added a substantial new appendix to address these points thoroughly.

- **Concerns about Evaluation Metrics and Validity**  **(LMAm: W5, Q5; Tsuo: W1; zXiG: W2, Comment 2; 69Yr: W4, W5, Q2):**  This was the most common concern. Reviewers questioned the metric's source, potential biases, and lack of validation.

  - **Our Addressing:**  We clarified that our metric is a **stricter, enhanced adaptation of the H-CoT evaluation protocol**. Specifically, we scale H-CoT's 0-5 LLM-based rating to a 0-100 **Alignment Score (AS)** , and then add our key enhancement: a **Keyword Suppression Score (KSS)**  that rewards responses free of refusal language. To provide full transparency and demonstrate its reliability, we have **added a new appendix** with illustrative examples for each score tier, showcasing how the metric consistently and accurately differentiates between various model behaviors.
- **Concerns about Novelty and Baselines**  **(LMAm: W2, Q1; zXiG: W1, Comment 1):**  Reviewers questioned the method's originality compared to other template-based or generative attacks and suggested more baselines.

  - **Our Addressing:**  We clarified that SAGE-CoT's core innovation lies in **cognitive process manipulation**—forcing the LRM to generate its own "Trojan horse" decoding template—which distinguishes it from prior art focused on input prompt optimization. Due to the significant inference time of LRMs, a full comparison against all suggested baselines was not feasible during the rebuttal period, but we have acknowledged them as crucial points of comparison and designated this as a **high-priority next step for our research**.
- **Concerns about Defense Evaluation (Tsuo: W3, Comment; 69Yr: W3, Q3):**  Reviewers noted that our defense suite was limited and requested evaluation against newer, stronger defenses.

  - **Our Addressing:**  We justified our initial defense selection based on the black-box threat model. More importantly, in direct response to reviewer feedback, we **conducted new experiments against the state-of-the-art Qwen3Guard defense**. The results provided a crucial finding: SAGE-CoT's performance was unaffected, while the H-CoT baseline's ASR dropped significantly, demonstrating our method's superior robustness. We have committed to evaluating against WildGuard as we continue to develop this work.
- **Concerns about Reproducibility and Clarity (LMAm: W1, W3, Q3, Q4; zXiG: Q2; 69Yr: W1, Q1):**  Reviewers encountered issues with unclear terminology and failed to reproduce our results.

  - **Our Addressing:**  We added a dedicated  **"Terminology Clarification"**  section to the paper. To resolve the reproducibility issues, we pointed to the **raw prompt files** in the supplementary materials (to avoid PDF formatting errors), specified the exact API settings required, and provided **successful execution logs** for key models as evidence.

We have made every effort to address all reviewer concerns point-by-point. The substantial new experiments (vs. Qwen3Guard) and the new appendix with metric examples were completed and added during the discussion period. We understand that reviewers may not have had sufficient time to fully review all new materials and update their assessments accordingly.

We are deeply grateful to the reviewers, AC, SAC, and PC for their dedicated efforts. Their insightful feedback has been instrumental in significantly strengthening our paper.

**Sincerely**,

**Authors**

---

### Meta-Review · Area_Chair_XAxs · 2026-01-09

**Summary:**

Reviewers agreed the paper addresses a timely safety issue and found the proposed SAGE-CoT attack effective and clearly presented. However, the rejection decision was driven by significant concerns about evaluation validity, baseline strength/fairness, novelty positioning, and evidentiary reliability. Multiple reviewers questioned whether the proposed GPT-4o–based metric (including keyword-suppression bonus) is sufficiently validated and whether it may advantage the method’s formatting/verbosity. Others argued the approach may be derivative of existing template-based and obfuscation jailbreaks and requested comparisons with stronger generative or optimization-based baselines (e.g., AutoDAN, GPTFuzzer, h4rm3l). Finally, reproducibility concerns—especially an inability to replicate key results on GPT-5 and Gemini—reduced confidence in the claimed robustness and practical impact.

**Reviewer Concerns:**

The rebuttal helped clarify terminology, added missing related work discussion, and provided more detail on metric construction, baseline implementation choices, and experimental settings. The authors also added new defense experiments (e.g., Qwen-Guard/Qwen3Guard), which strengthened the defense story. Nevertheless, key issues remain: the paper still lacks human validation of the evaluation metric, does not provide the requested comparisons against stronger baselines, and does not fully resolve independent concerns about reproducibility and real-world effectiveness. As a result, the committee still had insufficient confidence in the central claims to recommend acceptance.

**Reviewer Scores:**

Reviewer LMAm would likely remain at reject, as the rebuttal improved clarity but did not fully address metric validity and novelty concerns. Reviewer Tsuo might shift slightly upward due to the added defense experiments, but likely would remain borderline given continued concerns about metric bias and baseline fairness. Reviewer zXiG would likely stay near their initial marginal stance, since stronger baseline comparisons and human-aligned evaluation are still missing. Reviewer 69Yr would likely remain reject with high confidence, as the rebuttal did not provide sufficient independent evidence resolving reproducibility failures or strengthening claims about stealth and robustness.

---

### Decision · Program_Chairs · 2026-01-26

Reject